# Current Approaches to Osteoid Osteoma and Minimally Invasive Surgery—A Minireview and a Case Report

**DOI:** 10.3390/jcm11195806

**Published:** 2022-09-30

**Authors:** Jan Cerny, Jan Soukup, Sarka Cerna, Tomas Novotny

**Affiliations:** 1Department of Orthopaedics, University J.E. Purkinje and Masaryk Hospital, 401 13 Usti nad Labem, Czech Republic; 2Department of Rehabilitation and Sports Medicine, Second Faculty of Medicine, Charles University and University Hospital Motol, 150 06 Prague, Czech Republic; 3Department of Genetics, University J.E. Purkinje and Masaryk Hospital, 401 13 Usti nad Labem, Czech Republic

**Keywords:** osteoid osteoma, radiofrequency ablation, conventional C-arm guidance, minimally invasive surgery, benign bone tumors

## Abstract

Osteoid osteoma is a benign bone tumor typically affecting the long bones of the lower limbs in young male patients. The lesion can be asymptomatic but, in most cases, patients present with characteristic nocturnal pain that is very responsive to the administration of non-steroidal anti-inflammatory drugs. Although osteoid osteomas can regress spontaneously over time, surgical therapy is often indicated in cases of long-lasting resistant pain. Apart from a traditional open resection, the modalities of minimally invasive surgery, such as radiofrequency ablation or cryoablation, have gradually become the option of choice in most cases. The first part of this manuscript is a minireview of the contemporary literature on the pathogenesis, diagnosis, and current trends in the treatment of osteoid osteoma. The second part is a case report of our own experience with a conventional C-arm-guided radiofrequency ablation of an osteoid osteoma located in the femoral neck in an adolescent patient. The aim was to prove that, even when more sophisticated guiding devices (CT, O-arm, etc.) are not available, the safe and reliable ablation of the lesion using a C-arm is still possible even in hard-to-reach areas. The case was a success, with no perioperative or postoperative complications.

## 1. Introduction

Osteoid osteoma (OO) accounts for about 10–14% of all benign primary bone tumors [1]. It is a disease that predominantly affects male patients, typically aged 5–30 years [2]. The patients characteristically present with local pain with nocturnal maxima, which can be controlled well by the administration of non-steroidal anti-inflammatory drugs (NSAIDs) or salicylates. Other, less common symptoms may include growth disturbance, bone deformity, scoliosis, joint swelling and/or synovitis, restriction of the range of motion, or muscular contracture [2,3]. The most sensitive and most specific imaging method is computed tomography (CT), which, in the case of cortical-type osteoid osteoma, usually reveals the presence of a circular or oval hypodense osteolytic lesion, called a “nidus”. This area is lined with a hyperdense hyperostotic margin, sharply separated from the surrounding bone tissue [3]. OO can affect virtually any bone; however, it is mostly located in the metaphyses of the femur or tibia [2]. In the case of resistance to conservative treatment, surgical therapy is indicated. Operative treatment offers two main options; historically, the preferred method was an open wide en bloc resection. However, minimally invasive methods, such as radiofrequency ablation (RFA) or cryoablation, have gradually proven themselves to be simpler, far less invasive, and less complication-burdened options, and are already considered as the gold standard in the treatment of osteoid osteoma [2,3,4,5]. This manuscript presents a case report involving a young male patient with an OO located in the inner cortical surface of the right femoral neck, as well as a minireview of the recent literature dealing mainly with the pathogenesis, diagnosis, and current trends in the treatment of OO. While following current recommendations that prefer minimally invasive surgery (MIS), we decided to solve the case with RFA with the guidance of a conventional C-arm. We hope that this manuscript will provide the reader with an up-to-date overview of the current knowledge of OO, that it highlights the benefits of MIS in the treatment of such lesions, and that it shows that more sophisticated guidance modalities (O–arm, CT, Stealth, and 3D C-arm) can potentially be, in selected cases, substituted by a conventional C-arm.

## 2. Epidemiology

OOs make up approximately 2–3% of all primary bone tumors (benign and malignant lesions) [6,7,8]. The male-to-female incidence ratio is estimated to range between 2:1 and 4:1 [2]. The typical age of occurrence is 5–30 years [4]. OO most commonly affects the metadiaphyses of the long bones of the lower limbs, namely, the femur (50%) and the tibia (20%) [9]. The next most commonly affected areas are the spine (6–20%), specifically, the lumbar, cervical, and thoracic vertebrae, and the sacrum, in descending order of incidence. The posterior spinal column (spinous, transverse, and articular processes, and laminae, pedicles) is the most affected [9]. The incidence of OO in other areas is significantly less frequent; the upper extremities are affected in approximately 10% of cases [9,10,11]. Feet bones are involved in approximately 2–10% of cases. The involvement of the skull, facial bones, and other areas is extremely rare [9].

## 3. Classification

OOs can be classified based on several features: their localization in the body, the specific area of the bone, and their proximity to a joint. The classification schemes typically distinguish four types of OOs, according to their specific localization in the affected bone: intracortical (75%), subperiosteal (2.5%), endosteal (2.5%), and medullary (20%) [4]. Some authors [9] also distinguish a specific epiphyseal type. Based on proximity to a joint, we can divide OOs into extraarticular (extracapsular) and intraarticular (intracapsular) types, both with specific symptoms [11].

## 4. Origin and Pathophysiology

There are three main currents of opinion on the pathogenesis of OO. The most common opinion is that it is a benign bone tumor, while other authors suggest that local inflammation is to blame. Finally, alterations in the process of bone healing have been considered [4]. The neoplastic theory is supported by the presence of atypical osteoblasts and trabecular components in the lesion [4]. Furthermore, the central nidus is typically hypervascular due to pathologic angiogenesis [12]. On the other hand, OOs usually do not have the tendency to grow with time, whereas a spontaneous regression is not uncommon. The inflammation theory was proposed due to the presence of intracellular viral elements in osteoblasts, and because of the high production of inflammatory mediators, which are, most probably, also responsible for the generation of typical pain with nocturnal maxima [4,13]. The onset of the pain is attributed to the increased production of PGE2, prostacyclin (PGI2), and the overexpression of COX–2 [14,15,16]. Urinary concentrations of 2, 3–dinor–6–keto–PGF_1_α, which is a metabolite of extra-renally produced PGI2, are also usually elevated [15]. Based on these findings, two main mechanisms of prostaglandin-mediated pain in patients with OO have been proposed: (1) A prostaglandin-mediated increase in blood flow increases the intralesional pressure and thus overstimulates the nerve endings, and (2) prostaglandins irritate the nerve fibers directly. A combination of these two mechanisms is also possible [15]. The nocturnal predominance of pain is mainly attributed to the high concentration of unmyelinated nerve fibers [16]. Some authors [17] have stated that the creation of OO is an attempt at bone healing, but with no proof of preceding trauma, bone infarction, or infection. There are also studies that have reported the possible involvement of the 22q chromosome, which is a region that generally affects pathological cell proliferation in neoplasms [16]. In general, no causal etiological factor has been proven, and further research is needed [13].

## 5. Pathology

As stated above, osteoid osteomas have a relatively notable appearance. The central radiolucent area of the tumor, called the nidus, consists of osteoid-forming interconnected trabeculae of immature woven bone. The space among the trabeculae is typically filled with highly vascularized fibrous tissue (fibrovascular stroma), giant cells, and osteoclasts indicating active bone tissue turnover [4,13,14,18]. The nidus is surrounded by atypical osteoblasts with eccentric nuclei, small nucleoli, and abundant cytoplasm [14,15]. This central portion is sharply demarcated from the so-called reactive zone, which consists of sclerotic cortical or trabecular bone tissue [2,14]. The reactive zone tends to be more pronounced in the cortical type OO, as opposed to the medullar type. The whole lesion is typically smaller than 2 cm in diameter [2]. No malignant cells are present [15].

## 6. Diagnosis

### 6.1. Clinical Symptoms

The symptoms of OO may vary depending on the affected skeletal area and the proximity to a joint [5]. The pain is usually dull, but stabbing or knifelike sensations are also possible [4,19]. A limited range of motion and joint swelling with effusion due to synovitis can be the most evident symptoms of intraarticular or juxta-articular OOs [11,20,21]. These limitations can cause alterations in posture and gait stereotype [17]. In the case of the epiphyseal type of OO (found in children and adolescents), asymmetric growth resulting in length discrepancy and/or angular deformity of the limb is possible [21]. In subcutaneous areas, local erythema and tenderness is usual [15]. OOs of the spine can potentially cause painful scoliosis, or a torticollis in the cervical segment [4]. A neurological deficit due to extradural compression is more likely to occur in the case of spinal osteoblastoma [19,22]. OO localized in the upper extremity can often mimic various affections of the tendons and ligaments (e.g., epicondylitis, DeQuervain’s disease, calcifying tendonitis, etc.) [11]. The involvement of the hands often results in monoarticular arthritis and/or deformities such as clubbing or macrodactyly [4,11,23]. In patients with OOs localized in the foot, a typical nocturnal pain is more common than a deformity [24]. Sahin et al., in 2020, [9] reported that 73% of patients with OO localized in the lower extremities or in the pelvic bones had muscular atrophy. In addition, in 6% of the patients, the significance of the atrophy was responsible for posture and gait disorders.

### 6.2. Imaging Methods

The first imaging method used in the diagnostic algorithm is usually conventional radiography (CR) [12]. At least two orthogonal projections centered over the suspected area should be taken [4]. The typical image of the nidus is distinguished in about 85% of cases [12]. If difficulties are encountered with respect to distinguishing OOs with the use of CR (e.g., osteomas of the spine, femoral neck and small bones of the foot, or lesions smaller than 1 cm), other imaging modalities should be used [12,25]. Apart from the lesion itself, indirect skeletal consequences of OOs can also be observed, including scoliosis, bone overgrowth/deformity, or local osteopenia [11,25,26]. CT is considered to be the most sensitive imaging modality. Apart from a precise visualization of the hyperostotic margin of OOs, CT allows us to differentiate the mineralization of the nidus, which can be punctate, amorphous, or ring-like. The specific type of mineralization informs us mainly about the “age” of the lesion [15]. Another specific CT finding is the “vascular groove” sign, which represents the area where enlarged vessels enter the hypervascular nidus [12]. Despite previous views of its inferiority compared to CT, magnetic resonance imaging (MRI) has gradually become a very useful method in the diagnosis of OO. The lesion usually shows a difference in gadolinium uptake in the fibrovascular stroma of the nidus and in the hyperostotic rim. Typically, in a T2WI projection, the nidus shows an intermediately high intensity, in contrast with the rim, with a low gadolinium enhancement [12]. Cancellous OOs may be better visualized with MRI than with CT due to their lack of perinidal density alteration [27]. Dual-layer spectral CT is another useful method for the diagnosis of OO, as it combines the advantages of conventional CT (a precise determination of the bone lesion) and MRI (a quantification of bone marrow edema) [28]. Nuclear imaging offers three main methods for visualizing OOs: (1) 99mTc scintigraphy, with a reported detection sensitivity of nearly 100% (the typical finding is the double density sign, similar to the MRI picture); (2) single-photon-emission-computed tomography (SPECT), which is very useful in anatomically complex areas (e.g., the spine); and (3) 18–fluorodeoxyglucose positron-emission tomography (18–F–FDG–PET/CT), which can be used to assess the response to therapy [6].

### 6.3. Biopsy

Parmeggiani et al. (2021) [29] provided a thorough analysis of the current literature. They concluded that a histological verification preliminary to intervention is not a necessity when the OO has a characteristic appearance and when there are typical clinical symptoms. However, in cases of atypical presentation, a biopsy should be considered. Nevertheless, even with the current possibilities of CT navigation, a diagnostic pre-surgery biopsy is carried out in 36–73% of cases. In our department, we routinely draw biopsy samples intraoperatively to prove a diagnosis. A preoperative biopsy is usually not taken.

## 7. Therapy

### 7.1. Spontaneous Regression

Osteoid osteoma can be referred to as a self-limiting disease with a potential to regress spontaneously in anywhere between several months and 6–7 years after the onset of symptoms [2,30]. However, a spontaneous regression is not very common (e.g., Aiba et al. [30] reported a remission in only 2 out of 11 patients).

### 7.2. Medical Therapy

Another modality of conservative treatment is the administration of salicylates and/or NSAIDs. The effect of such medication is attributed to the inhibition of PGI2 and PG2 secretion and COX–2 expression. Among the salicylates, aspirin was the most used drug in the studied literature. Carpintero-Benitez et al. (2004) [31] found a better response to therapy with Rofecoxib (a selective COX–2 inhibitor) as opposed to other NSAIDs. The initial effect of NSAIDs and/or salicylates is usually immediate, but a long-lasting remission of pain is not certain. The efficacy of NSAIDs and/or aspirin varies from study to study and is usually about 30–90% [32,33,34]. Goto et al. [35] found that 8 out of 12 patients in his study were pain-free after discontinuing NSAIDs for a period of 2–36 months. Bisphosphonates (pamidronate and zoledronate) can also relieve pain [36]. Although conservative treatments can be effective in many patients, the side effects of NSAIDs and salicylates must be considered thoroughly, especially in the age group of patients that are most often affected by OO. The most common side effects are refractory pain, sleep deprivation, and gastrointestinal irritation [2].

### 7.3. Open Surgery

Surgical therapy is usually indicated in cases when conservative treatment is no longer effective, or if the dosage of the drugs must be systematically increased to maintain the remission of pain [5,35]. Historically, open en bloc resection was the method of choice. The effectiveness of an open procedure is reported to be 88–100% [4]. The aim is to excise the entire nidus, the surrounding hyperostotic rim, and to carry out a local bone curettage [4]. This is often easier in theory than in practice, because localizing the nidus is not always simple. Therefore, extensive bone resections, which can significantly compromise the stability of the affected bone, sometimes have to be performed [2]. Such situations subsequently require an additional internal fixation and/or bone grafting, to prevent iatrogenic/pathological fracture [2]. Intraoperative CT/O-arm/C-arm/99mTC guidance to some extent reduces the risk of missing the exact position of the lesion [4].

### 7.4. Minimally Invasive Approaches

Several minimally invasive approaches to the treatment of OO have been proposed, including radiofrequency ablation (RFA), cryoablation, (CT-guided) percutaneous excision, microwave ablation (MWA), magnetic resonance-guided focused ultrasound (MRg–FUS), and interstitial laser ablation (ILA) [4,28]. RFA is still considered to be the gold standard for MIS approaches in the treatment of OO. Typically, a probe is inserted into the lesion (under radiographic guidance) and the ablation program is initiated. The probe is usually heated to a certain temperature, depending on the manufacturer (70–90 °C). The temperature determines the time for which the probe is inserted (generally 5–6 min). In RFA, guidance can be carried out through several devices, which provide different levels of accuracy (CT, O-arm, 3D Iso-C C-arm, stealth navigation, or conventional C-arm, which was used in our case study). RFA is generally a safe method, which can be repeated in the case of a relapse. However, some complications can be encountered, including site burns, local infections, neural damage, or a probe fracture [27,37,38]. In spinal OOs, intralesional extracapsular excision is usually still reported to be a safer method than RFA due to the proximity of neural structures [39]. The main benefit of cryoablation is the possibility to visualize the course of the procedure (the gradual creation of a hypodense area visible on CT—the so-called “ice ball sign”) in real time. This allows the surgeon to terminate the procedure precisely when the lesion is sufficiently ablated, which prevents any significant collateral damage, and is, therefore, safer even in the proximity of neural structures [29]. The success rate is reported to be about 96–100% [40,41]. MWA is a promising method, but a feared potential complication is a secondary fracture in the affected area, as the temperatures are higher and are reached more quickly than with RFA. The probes used in MWA do not have to be positioned precisely in the center of the nidus; however, the overall exposed area can, therefore, be larger [29]. ILA should also be considered, as it is cheaper than RFA, and it does not involve the passage of an electric current through the patient’s body [29]. Its success rates are also respectable, usually ranging between 94 and 100% [42,43]. MRg-FUS is a new and promising method, as it is completely non-invasive and radiation-free. The technique uses the conversion of mechanical energy to heat. However, it is contraindicated in hard-to-reach areas, and it does not allow for a biopsy to be taken. The success rate of this method ranges from 87–100% [29,44,45].

## 8. Differential Diagnosis

Many afflictions can mimic OO, especially based on their size, location, clinical symptoms, and pathology [7]. Firstly, we should rule out the possibility of osteoblastoma, which is usually larger than 2 cm in diameter and its growth can be locally aggressive, tends to be less painful, and this pain is typically less responsive to NSAID and/or salicylates [12]. Focal osteomyelitis or intraosseous abscesses are also typically larger, and they do not show a sharp separation from the surrounding bone tissue as in OO [12]. Chondroblastoma is typical for its epiphyseal location and irregular margin. Sometimes, especially in cases of unspecific bone marrow edema, we should rule out the possibility of a stress fracture. The absence of the nidus and gradual spontaneous regression after a conservative treatment are sufficient in a differential diagnosis. A glomus tumor of the phalanges can have a similar appearance to OO; however, the gadolinium enhancement upon MRI usually tends to be more homogenous and it lacks the perinidal hypo-intensity [12]. Crystal deposition diseases should be mentioned as well, as they can cause bone marrow edema; however, the more frequent affection of tendons and ligaments and the higher age of the affected patients (typically older than 30 years of age) usually facilitate the correct diagnosis [12]. Perthe’s disease should also be excluded; however, the typical radiographic changes, such as the fragmentation of the femoral head, are very specific to this condition.

Finally, considering the typical clinical presentation of OO and its characteristic appearance in imaging methods, a correct diagnosis is usually not very problematic. The diagnostic pathway as described in the section “Diagnosis” should provide a sufficient confirmation of OO. In cases of atypical signs and symptoms, a preoperative biopsy should be considered.

## 9. Case History

Our patient is a 16-year-old Caucasian male, who initially presented to an orthopedic surgeon with a 4-month lasting pain in the right hip extending to the anterior surface of the right thigh. The patient had a history of developmental dysplasia of the hips, which was successfully treated with a Pavlík harness, as the patient had previously been pain-free, despite regular high-intensity sports activities. An ultrasound of the right hip joint was performed with the finding of synovitis. Due to the ongoing pain and the atypical symptoms, an MRI of the hip was taken, and it showed an approximately 1 × 1 cm large ovoid lesion on the internal surface of the medial cortex of the neck of the right femur (just above the Adams arch), which was suspected to be an osteoid osteoma. After this examination, the patient was referred to our department. We found severe atrophy of the anterior right femoral muscle mass, together with a slight (approximately 10–degree) flexion deficit in the right hip joint.

To prove the diagnosis of OO, a CT scan of the affected area was taken. The condition was consulted with a neurologist, who found decreased reflexivity in the L2–L4 myotomes, and, therefore, recommended a lumbar spine MRI to rule out any spinal lesions (e.g., a tumor). No such affection was found. Even though our diagnostic algorithm was not standard (among other things, due to the fact that the patient was referred to our workplace only after the MRI was performed), we arranged the individual diagnostic steps and the Figure 1, Figure 2 and Figure 3 (i.e., clinical examination–CT–MRI) in the correct order for better clarity.

Considering the findings and the persistent pain even after the regular administration of NSAIDs, a surgical solution was indicated in accordance with the patient and his legal representatives. Reaching the osteoma would be very challenging through a standard open approach, and the high level of invasiveness of such a procedure in a teenage patient would be unfavorable at the very least. Therefore, we turned to a more appropriate alternative in the form of a minimally invasive RFA. We implemented the OsteoCool^TM^ Bone Radio Frequency (RF) Ablation System (Sofamor Danek/Medtronic, Dublin, Ireland), which consists of an RF generator, a peristaltic pump, a connector hub with two channels for the RFA probes with an integrated cooling system for the active tips, and two channels for optional thermocouples [46]. The surgeon can intraoperatively choose the appropriate size of the active tip of the probe with different ablation volumes and times, depending on the size of the lesion. First, after a 2 cm skin incision above the greater trochanter, a Kirschner wire was inserted via a lateral pertrochanteric approach to the lateral hyperostotic margin of the osteoma. The insertion was carried out under direct radiographic control, using conventional C arm guidance (Figure 4).

The correct positioning of the instruments was always controlled in two perpendicular projections (anteroposterior and lateral). Subsequently, we inserted an over-the-wire working cannula to draw a core biopsy sample for a further histologic examination (Figure 5).

Then, after an additional disruption of the nidus with a drill, we commenced with inserting the RFA probe (Figure 6).

We used a 7 mm probe tip heated to 70 °C for 6.5 min. After finishing the ablation program, we extracted the probe with the working cannula and finished the procedure with two absorbable stitches of the skin (Figure 7).

There were no intraoperative complications. After the procedure, the patient started postoperative rehabilitation in the inpatient center of our department. He was completely pain-free and was able to stand and walk with two crutches on the first post-op day. No postoperative complications emerged, and the patient was discharged on the third day after the procedure, able to fully load the operated limb. At the 6-week post-op check-up, an MRI scan was taken, which showed the complete removal of the OO.

A histological examination proved that the lesion was indeed an osteoid osteoma. Further checkups have already been carried out at 3 months and at 6 months post-op. The patient has returned to his regular sports routine, and the musculature on his right lower extremity is gradually regenerating (Figure 8). At the 6-month post-op check-up, we took a long-format standing radiograph, which showed a slight valgus orientation of the proximal femurs; however, the stability of the right hip joint was not jeopardized even after our procedure (Figure 9). An MRI scan was also taken, and it proved the complete removal of the OO as well as the healing of the working canal (Figure 10). A further check-up is planned at one-year post-op.

## 10. Discussion

Our case was interesting due to the fact that we used conventional 2D C-arm radiographic guidance for targeting the OO, which has not been previously reported in the studied literature (other authors used 3D C-arm imaging [37,47]. This was partially due to a mechanical failure of the O-arm device that we had originally planned to use. However, considering the many years of our experience with C-arm guidance for treating various spinal lesions (e.g., vertebral fractures, metastases, hemangiomas, etc.), we determined this type of navigation to be safe. No complications were encountered during the procedure. On the contrary, in terms of accuracy, we cannot argue with the fact that CT/O-arm/3D C-arm types of guidance are superior methods to the conventional 2D C-arm. The possible risks of the “imperfect” 3D visualization of the lesion are mainly missing the center of the tumor and thus its incomplete removal, which can lead to greater risk of a relapse. A higher radiation exposure can also occur, mainly when localizing the lesion in separate planes is difficult, and so more radiographs must be taken. Many authors have stated that RFA should still be considered the gold standard in the minimally invasive treatment of OO. The method has been thoroughly tested through the years, and it has proven itself to be a highly successful, cost-effective method, with few complications [48,49]. The method’s cost-effectiveness is supported not only by the cost of the procedure itself, but also by the shorter hospital stay [48]. On the other hand, given that teenagers are the most common clientele for the treatment for OO, reducing radiation exposure is an important topic. It will, therefore, be interesting to follow the further development of newer methods, such as MRg – FUS, which not only allows us to eliminate radiation exposure but also does not require invasive entry into the integrity of the body. Another example of an interesting approach may be the use of cryoablation, which, unlike RFA, allows for the live monitoring of the procedure [29]. However, not all cases and types of OOs can be treated with the newer methods. Another topic for discussion is the position of the traditional open surgery among contemporary trends in the treatment of OO. Our experience suggests that it should certainly not be neglected. It has advantages in cases of superficial and well-differentiated osteomas, where open surgery can potentially be less invasive than some MIS methods as there is a lower risk of, e.g., a potential skin burn. Hamdi et al. (2014) [50] proposed that open surgery is the option of choice in treating OOs in and around the hand. However, other authors presented large groups of patients with the conclusion that MIS methods should be generally preferred not only due to their lower invasiveness, but also because of their use resulting in a lower recurrence rate (18, 1% in open surgery and 15, 3% in RFA group) and the lack of requirement for additional casting or bracing after the procedure [51,52]. On the contrary, Rosenthal et al. (1998) [53] found that there is no significant difference in the recurrence rates between RFA and open surgery, or even that RFA’s rates are slightly higher (9% in open surgery; 13% in RFA). Spinal OOs are also controversial, as some authors still strongly recommend giving preference to standard open surgery because of the proximity of neural structures [12,54]. However, there have already been studies that have proved that even in the spine RFA can be advantageous [51]. Finally, it is necessary to point out that all methods have their limitations, and that a specific treatment should be set up individually to ensure a safe and reliable procedure.

## 11. Conclusions

Osteoid osteoma is a common benign bone tumor and should, therefore, always be taken into consideration in cases when young patients report nocturnal limb/joint pain. A wide range of minimally invasive methods are available, covering the majority of operative indications, and we should approach them (especially RFA) as the treatment of choice. The main conclusion of this manuscript is that RFA in the treatment of OO is possible even when a 3D visualization of the lesion is omitted. Open surgery is always an option (even preferable in, e.g., spinal OOs), but despite limited technical possibilities, MIS methods are usually safely feasible.

## Figures and Tables

**Figure 1 jcm-11-05806-f001:**
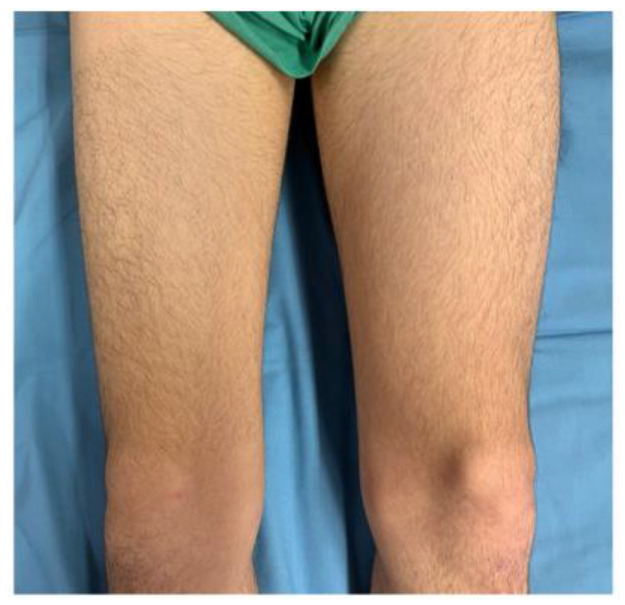
Notable atrophy of the anterior thigh muscle mass of the right lower extremity as a consequence of its long-term load relief due to peristent pain.

**Figure 2 jcm-11-05806-f002:**
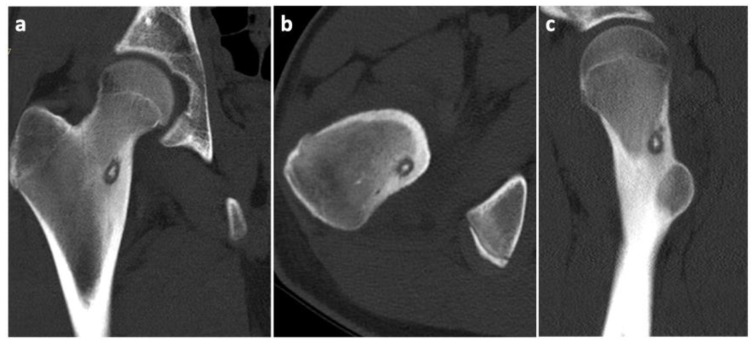
CT scans of the osteoid osteoma in different projections. (**a**) Frontal view; (**b**) Transverse view; (**c**) Sagittal view.

**Figure 3 jcm-11-05806-f003:**
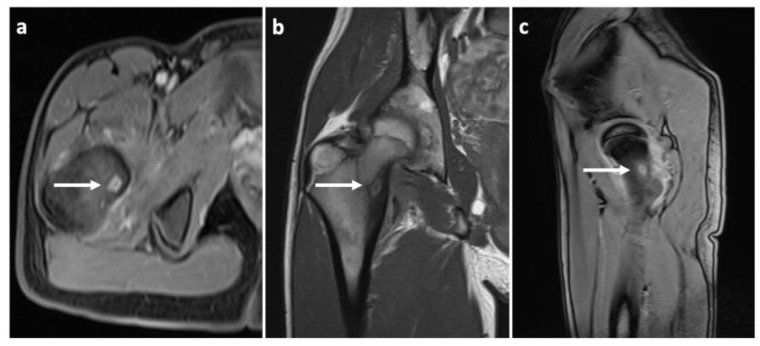
Different MRI (magnetic resonance imaging) views with white arrows indicating the location of the osteoid osteoma. (**a**) T1 vibe dixon transverse projection; (**b**) T1 TSE (turbo spin echo sequence) frontal projection; (**c**) T1 vibe dixon sagittal projection.

**Figure 4 jcm-11-05806-f004:**
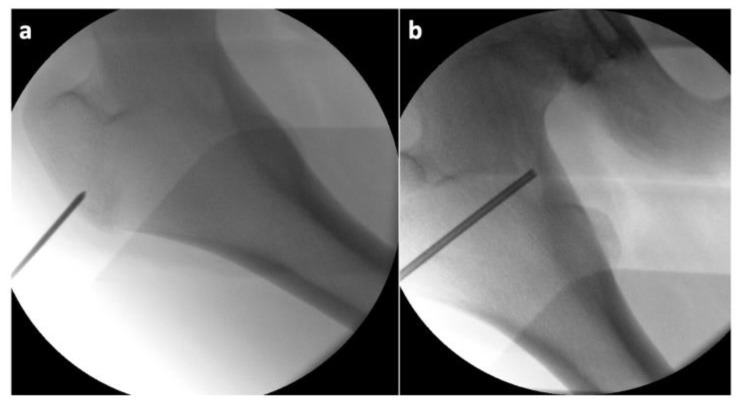
Intraoperative X-ray of the proximal femur showing: (**a**) the insertion point of the Kirschner wire; (**b**) the position of the working cannula in the osteoma (the slightly brighter irregular oval area).

**Figure 5 jcm-11-05806-f005:**
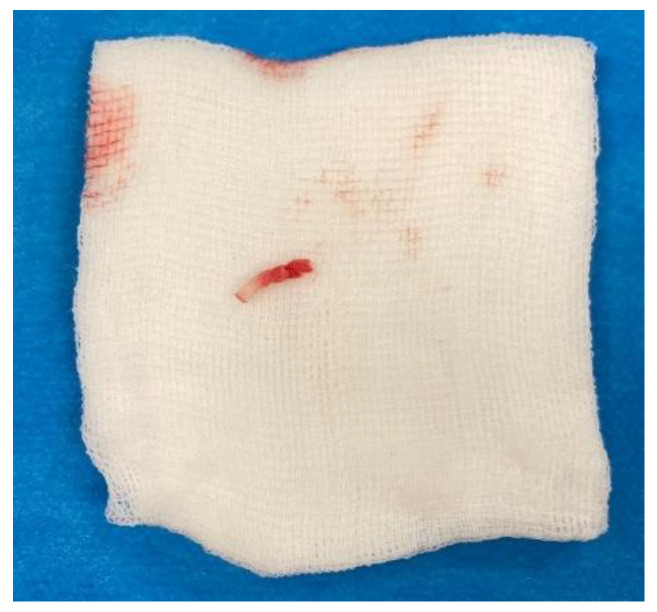
A bioptic sample of the osteoma.

**Figure 6 jcm-11-05806-f006:**
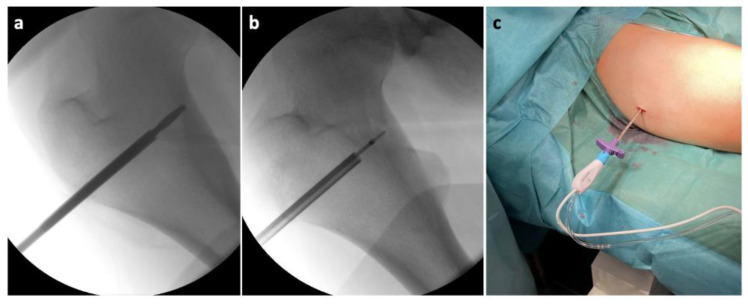
Intraoperative X-ray of the proximal femur showing: (**a**) disruption of the nidus with a drill; (**b**) positioning of the RFA (Radiofrequency abaltion) probe in the nidus; (**c**) the RFA probe inserted in the working cannula.

**Figure 7 jcm-11-05806-f007:**
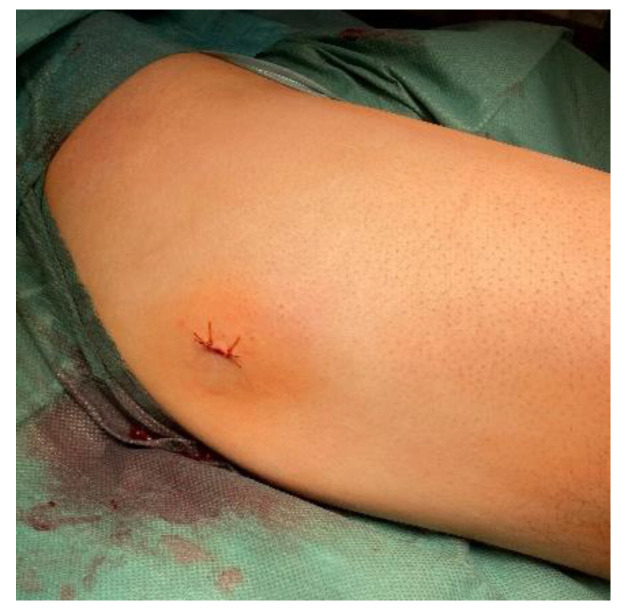
Postoperative condition in the operating room: an approximately 2-centimeter incision.

**Figure 8 jcm-11-05806-f008:**
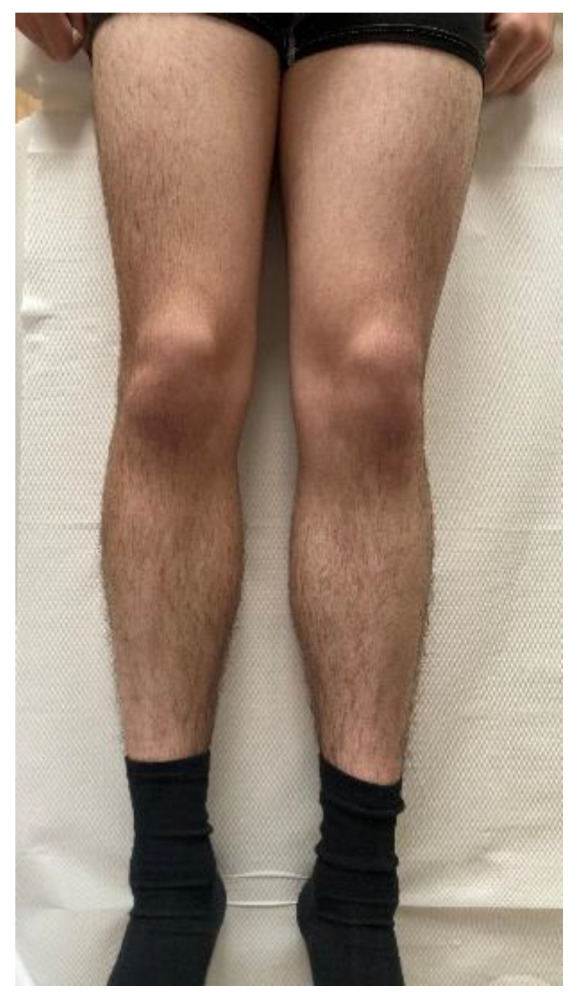
Proportional regeneration of the right femoral anterior muscle mass at 6 months post-op.

**Figure 9 jcm-11-05806-f009:**
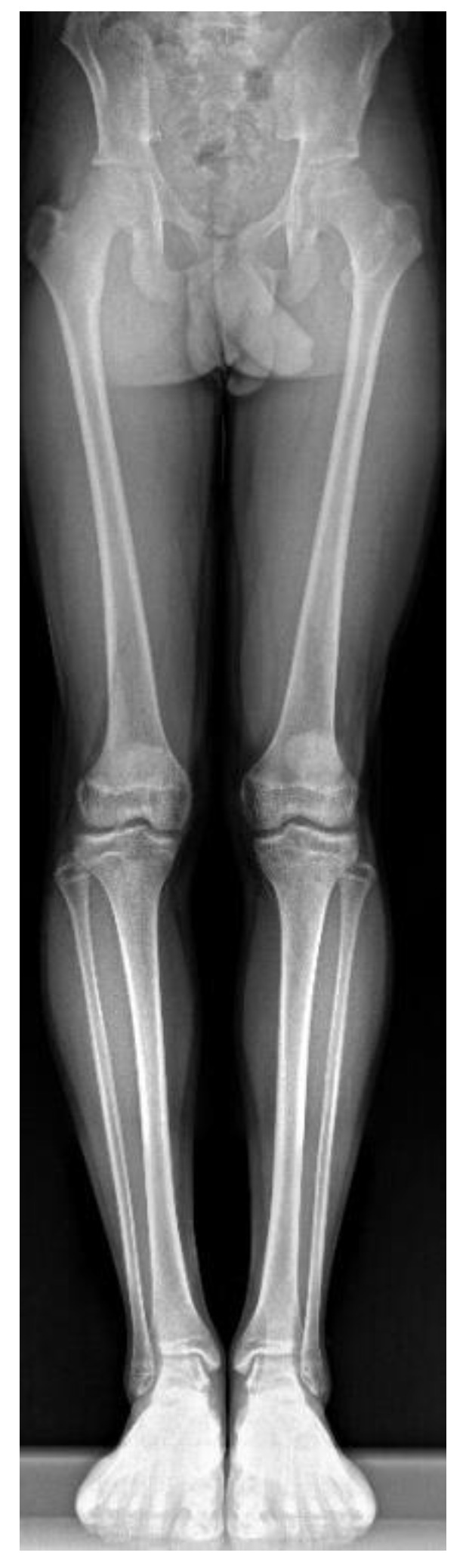
Long-format X-ray of the lower limbs showing a slight bilateral acetabular dysplasia of the hips due to a history of DDH (Developmental dysplasia of the hips). The centrum–collum–diaphysis angles are 132° on the right side and 134° on the left side.

**Figure 10 jcm-11-05806-f010:**
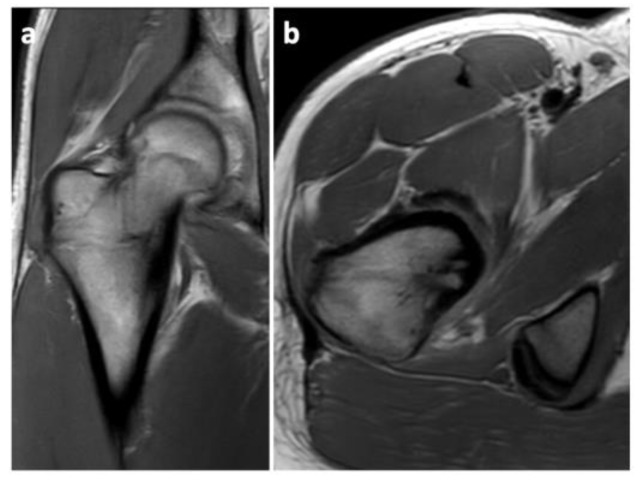
A 6-month post-op MRI scan showing no signs of relapse of the OO. (**a**) Frontal projection; (**b**) Transverse projection.

## Data Availability

The data that support the findings of this study are available from the corresponding author upon reasonable request.

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
