# Peer review of "Current Approaches to Osteoid Osteoma and Minimally Invasive Surgery—A Minireview and a Case Report"

_jcm, 2022, doi:10.3390/jcm11195806_

Round 1

Reviewer 1 Report (Previous Reviewer 3)

There are three questions require explanation 

1. Standing AP radiograph shows suspected coxa valga. Did the authors measured the angle between the femoral neck and the femoral shaft?

2. The acetbulae look dysplastic, are the authors aware of the incomplete coverage of the femoral head (instability of the hip)?

3. Was there any history of developmental dysplasia of the hip (DDH)?

Author Response

Enclosed 

Reviewer 2 Report (New Reviewer)

Thank you very much for the chance to review this manuscript. The authors provided a non-systematic literature review regarding the pathogenesis, diagnosis, and treatment of osteoid osteoma. Further, they report a case treated under c-arm guidance.

Unfortunately, I don't see any significant contribution to the existing literature. Background information provided regarding osteoid osteoma was already provided by recent reviews, e.g.,:
https://pubmed.ncbi.nlm.nih.gov/34182465/
https://www.ncbi.nlm.nih.gov/pmc/articles/PMC8286494/

The case presented seems also not novel. C-arm usage in osteoid osteoma was already reported by previous work groups in contrast to the statement of the authors ("Our case was interesting in the way that we used conventional C–arm radiographic guidance for targeting the OO, which has not been previously reported in the studied literature.").

The studies are for example:
https://pubmed.ncbi.nlm.nih.gov/26415989/
https://pubmed.ncbi.nlm.nih.gov/27384729/

I highly appreciate the efforts the authors made in preparing this work. The work is presented in an appropriate way. However, I think this study is redundant to the available evidence 

Author Response

Firstly, on behalf of all the co-authors, we would like to thank the reviewer for taking their valuable time to elaborate on our manuscript. The comments of the reviewer were very apt and allowed us to optimise the structure of our work. We tried our best to meet the requirements in individual points and sincerely hope, that our manuscript is acceptable in its present form for publication in your highly esteemed journal. All revisions are indicated in red font in text (visible corrections form).

Point 1:

Thank you very much for the chance to review this manuscript. The authors provided a non-systematic literature review regarding the pathogenesis, diagnosis, and treatment of osteoid osteoma. Further, they report a case treated under c-arm guidance.

Unfortunately, I don't see any significant contribution to the existing literature. Background information provided regarding osteoid osteoma was already provided by recent reviews, e.g.,:
https://pubmed.ncbi.nlm.nih.gov/34182465/
https://www.ncbi.nlm.nih.gov/pmc/articles/PMC8286494/

The case presented seems also not novel. C-arm usage in osteoid osteoma was already reported by previous work groups in contrast to the statement of the authors ("Our case was interesting in the way that we used conventional C–arm radiographic guidance for targeting the OO, which has not been previously reported in the studied literature.").

The studies are for example:
https://pubmed.ncbi.nlm.nih.gov/26415989/
https://pubmed.ncbi.nlm.nih.gov/27384729/

I highly appreciate the efforts the authors made in preparing this work. The work is presented in an appropriate way. However, I think this study is redundant to the available evidence 

Response 1:

Thank you for your comments. We are aware that the main topic of this manuscript has been addressed before, however, we hoped that an updated review of the literature would be useful in day – to – day clinical practice of the readers. Furthermore, the two articles related to perioperative navigation were both reviews, where 3D Iso C – arm was used. In our case, we had to rely on conventional 2D C – arm and thus regularly change projections (AP and lateral) intraoperatively to ensure correct positioning of the instruments. Despite the fact that we would have preferred a more sophisticated guidance device, we wanted to share our experience with this method and prove that intraoperative 3D visualization of osteoid osteoma is not always an absolute necessity.

Reviewer 3 Report (New Reviewer)

Overall, this is a well written review of OOs and an interesting case report of using a C-arm for ablation of an OO in the femoral neck. 

The review section mainly is well written - here I just have some minor comments. The case report needs some editing mainly in terms of the diagnostic storyline and the conclusion / clinical use case. Please find my comments attached:

-Explain clinical use cases —> Usually O-arm guidance for intervention? —> But if no O-arm or CT is available (that’s the described use case here), how is it diagnosed?

-Better differentiate between the sections “Origin and pathogenesis”, “pathology and pathophysiology” and “Imaging methods”. Those are partially mixed up.

-Differential diagnosis is too short. Describe pathways if diagnosis is unsure and include possible DDs and how those are differentiated.

-Maybe include dual-layer spectral CT in diagnosis-section (Gassert FT, Hammel J, Hofmann FC, Neumann J, von Schacky CE, Gassert FG, Pfeiffer D, Pfeiffer F, Makowski MR, Woertler K, Gersing AS, Schwaiger BJ. Detection of Bone Marrow Edema in Patients with Osteoid Osteoma Using Three-Material Decomposition with Dual-Layer Spectral CT. Diagnostics (Basel). 2021 May 26;11(6):953. doi: 10.3390/diagnostics11060953. PMID: 34073416; PMCID: PMC8227561.)

-Figure 1: Low image quality.

-Regarding Figures: Go by the clinical diagnostic pathway: Start with clinical images. Then X-ray (should be available as you propose ablation in areas where no sophisticated imaging methods are available). Then CT and then MRI. —> Make this a storyline to follow.

-Discussion is missing references (only three references in the entire discussion). E.g. method is cost-effective —> Is that an assumption or proven by studies?

-The authors present a case which was mainly based on a failure of the initially planned method (O-arm). Nevertheless I do not fully understand which conclusion is derived from this and for what clinical scenario —> To be used in countries which lack O-arms or just in cases where the O-arm is not working? 

-Risks which occur when using an C-arm instead of an O-arm should be addressed.

Round 2

Reviewer 2 Report (New Reviewer)

Thank you very much for your reply. The manuscript is well written and researched.

This manuscript is a resubmission of an earlier submission. The following is a list of the peer review reports and author responses from that submission.

Round 1

Reviewer 1 Report

The authors present a well-writen mini-review of pathophysiology and treatment of osteoid osteoma. Also the report a case of a young patient with an osteoid osteoma treated by minimally invasive surgery and radiofrequency ablation. They used a conventional C-arm radiographic guidance toe locate the nidus. The authors present this method of guidance as an alternative to CT-scans or O-arm  guidance. Most osteoid osteomas are not visible on plain x-rays and even with CT guidance the accuracy of treatment is around 80% ( Nijland H, Gerbers JG, Bulstra SK, Overbosch J, Stevens M, Jutte PC (2017) Evaluation of accuracy and precision of CT-guidance in Radiofrequency Ablation for osteoid osteoma in 86 patients. PLoS ONE 12(4): e0169171. ) Therefore using an standard  C arm might be feasible in only selected cases. 

Reviewer 2 Report

Thank you for the opportunity to review this article. The article describes a case of osteoid osteoma of the femur treated with RFA using C-arm to localize the lesion. The case is interesting. The article, in particular the state-of-the-art review on osteoid osteoma, is excellently written. However, in my opinion, the publication of case reports should give priority to cases of rare diseases or those with unconventional presentation, new treatments or complications that cannot be predicted. This facilitates the emerging and consultation of new and unexpected elements in the medical literature. In this case, unfortunately, neither the pathology, nor the treatment, nor the complication describe any exceptional situation. In fact, I think that the use of a C-arm to localize osteoid osteomas is not advisable as it is not possible to localize the lesion in too much cases to suggest its use. Moreover, there are no clear elements that can help to identify which case can be manage with this method. An extensively use of C-arm would likely lead to an increase in recurrences, which are already numerous even with more accurate detection methods. I therefore think that the review adds nothing to the literature and the report describes an unsuitable practice in my opinion. In conclusion, unfortunately, although I congratulate the Authors on the high quality of the paper, 

Reviewer 3 Report

Authors, please provide a full length standing AP radiograph for both lower limbs to visualize the pelvis, knees and ankle joints in one film in order assess the alignment and knee stability.